# Ecological Connectivity of Vicuña (*Vicugna vicugna*) in a Remote Area of Chile and Conservation Implications

Cristina Mata [1,2,*], Benito A. González [3], Denise S. Donoso [4], Nicolás Fuentes-Allende [5,6], Cristián F. Estades [3] and Juan E. Malo [1,2]

1   Centro de Investigación en Biodiversidad y Cambio Global (CIBC-UAM), Universidad Autónoma de Madrid, C. Darwin, 2, 28049 Madrid, Spain; je.malo@uam.es
2   Terrestrial Ecology Group-TEG, Departamento de Ecología, Universidad Autónoma de Madrid, C. Darwin, 2, 28049 Madrid, Spain
3   Laboratorio de Ecología de Vida Silvestre, Facultad de Ciencias Forestales y de la Conservación de la Naturaleza, Universidad de Chile, Av. Santa Rosa 11315, La Pintana, Santiago 8330015, Chile; bengonza@uchile.cl (B.A.G.); cestades@uchile.cl (C.F.E.)
4   Independent Researcher, Santiago 8301268, Chile; dsdonoso@gmail.com
5   Fundación Sudamérica Diversa, Panguipulli 5210205, Chile; fuente.nicolas@gmail.com
6   INIA Ururi, Instituto de Investigaciones Agropecuarias, Arica 1001219, Chile
*   Correspondence: cristina.mata@uam.es; Tel.: +34-91-497-8011

**Abstract:** Ecological connectivity is key for the long-term viability of species and is necessary when facing disturbance or global change, and geospatial analysis tools are key to exploring it with conservation aims. The vicuña is an ungulate endemic from South American highlands that faced extinction risk fifty years ago and is now slowly expanding and increasing in numbers. At present, it has a patchy distribution that may partially reflect connectivity limitations, an issue which can be key for its conservation under climate change. We developed a habitat suitability model using MaxEnt and location data of vicuñas in the Tarapacá region (Northern Chile), combined with spatial layers derived from remotely sensed imagery. We then used these results as the basis for a cost surface layer, and we examined habitat connectivity using least-cost and graph theory methods. Results showed the relevance for the species of habitat patches in the Southern part of the study area, out of protected areas, and the fact that ecological connectivity relies mainly on the intra-patch and flux components. These results should guide conservation actions for the species in the area and exemplify the relevance of remote sensing and geospatial models in the study of remote areas.

**Keywords:** circuitscape; connectivity conservation; fragmentation; graph theory; modeling; protected areas; spatial conservation planning; ungulate





## 1. Introduction

Ecological connectivity is the unimpeded movement of species and the flow of natural processes that sustain life on Earth [1], and it strongly depends on landscape structure (e.g., amounts and arrangement of types of landcover) and the response of each species to it. This response is species-specific, and it is rooted in habitat selection, movement capacity and dispersal distance [2]. The maintenance and improvement of connectivity is key to ensuring the long-term viability of species, as it helps maintain the genetic and demographic flows that enable population persistence in the face of local disturbances and changes in land use and climate [3–6]. Therefore, the maintenance and/or recovery of connectivity may be key for species threatened with extinction, and conservation planning for them should pay attention to the issue to achieve a favorable status for the species backed on natural processes [7,8]. This is particularly important for species that have been affected by retaliatory hunting, overexploitation or illegal hunting, such as large carnivores and ungulates surviving in small numbers with a patchy distribution [9–12].

The vicuña (*Vicugna vicugna*) is a wild camelid native to South America, which faced extinction risk a few decades ago. Its current distribution covers the central Andes of Peru, Bolivia, Argentina and Chile between 9°30′ and 29°30′ S and between 2800 and 4800 m asl [13]. In fact, its abundance declined from approximately two million individuals to only a few thousand from the European conquest until the 1960s, due to hunting for their fine fiber [14,15]. At the time of the lowest population size, only 6000 to 10,000 vicuñas remained dispersed in small groups, with most local populations extirpated and only 17% of their original area occupied [16–19]. This situation was reversed thanks to the establishment of national parks and reserves, the signing of the Vicuña Convention and the provision of funds by international NGOs to finance its protection. These measures allowed a significant increase in the vicuña population in five decades to roughly 460,000–520,000 individuals [20]. They have also slowly recolonized areas where they had been decimated, and there is currently a mosaic of areas with high abundance of the species, areas of low abundance and vast extensions with no records of vicuñas [20]. Thus, factors explaining this current pattern of heterogeneous presence of the vicuña in the landscape are unknown and they could partially be shaped by connectivity. On the one hand, it could be related to the heterogeneous distribution of habitat patches, and on the other hand, it could be their rather sedentary behavior, with daily and seasonal movements of very short distance [21–23], which would limit the expansion of population nuclei at broader landscape scales.

The vicuña also has a discontinuous presence in Chile. From the border with Peru in the north to 30° S at its southern limit, the protected areas of Lauca National Park, Las Vicuñas National Reserve and Isluga Volcano National Park are home to the largest vicuña populations in the extreme north of Chile. To the south, protected areas are few and scattered, and animal density is low with populations found outside of parks and reserves. In all these situations, the vicuña is subject to threats such as competition with domestic livestock [24], poaching [25] and mining activities [26], which could also affect intra- and inter-population connectivity. Specifically in the Tarapacá region, populations of the species inhabit between 3700 and 4800 m above sea level [24] under extreme environmental conditions typical of the altiplano, such as low precipitation, high solar radiation and thermal amplitude [27]. Their diet is mainly composed of perennial grasses and shrubs [28], which determines their occupation of both azonal swampy 'bofedales' and zonal plant formations [29]. Although there is no region-wide population monitoring, it is estimated that, for the Tarapacá altiplano, the density varies between 0 and 21 vicuñas/km$^2$, with the most common density being <2 vicuñas/km$^2$ [30]. The area reflects a slow recovery in numbers and distribution of the species [31]. Since the environment where the vicuña inhabits is itself heterogeneous [24], this recovery in distribution would be favored by a landscape with high ecological connectivity, for which the identification of habitat patches available at distances that allow vicuñas to move between them, and the resistance of the landscape to do so, would be a relevant step for the management of the species.

In this context, the main objectives of this study are to determine the main environmental factors related to the presence of the species, and to identify the habitat patches that are relevant for ecological connectivity in the Tarapacá region. This would provide spatially explicit information for the design of conservation strategies for the species in the region, as well as an example for other areas of vicuña distribution and a methodological framework for other species in equivalent situations.

## 2. Materials and Methods

### 2.1. Study Area

The study was carried out in the administrative region of Tarapacá, Chile. The area comprises a total surface of ~11,200 km$^2$ (Figure 1), between 3700 and 4900 m asl. It is a predominantly arid zone, highly conditioned by the altitude and climate, where the main plant associations are the bofedal (*Oxychloe andina*), pajonal or coironal steppe (*Festuca ortophylla—Deyeuxia breviaristata*) and tolar scrub (*Festuca ortophylla—Parastrephia lucida o Parastrephia quadrangularis*), with scattered spots of llaretal cushion-like formations

(*Azorella compacta—Parastrephia quadrangularis*) and queñoal open woods (*Polylepis tomentella* (=*Polylepis tarapacana*)-*Festuca* sp.) [32].

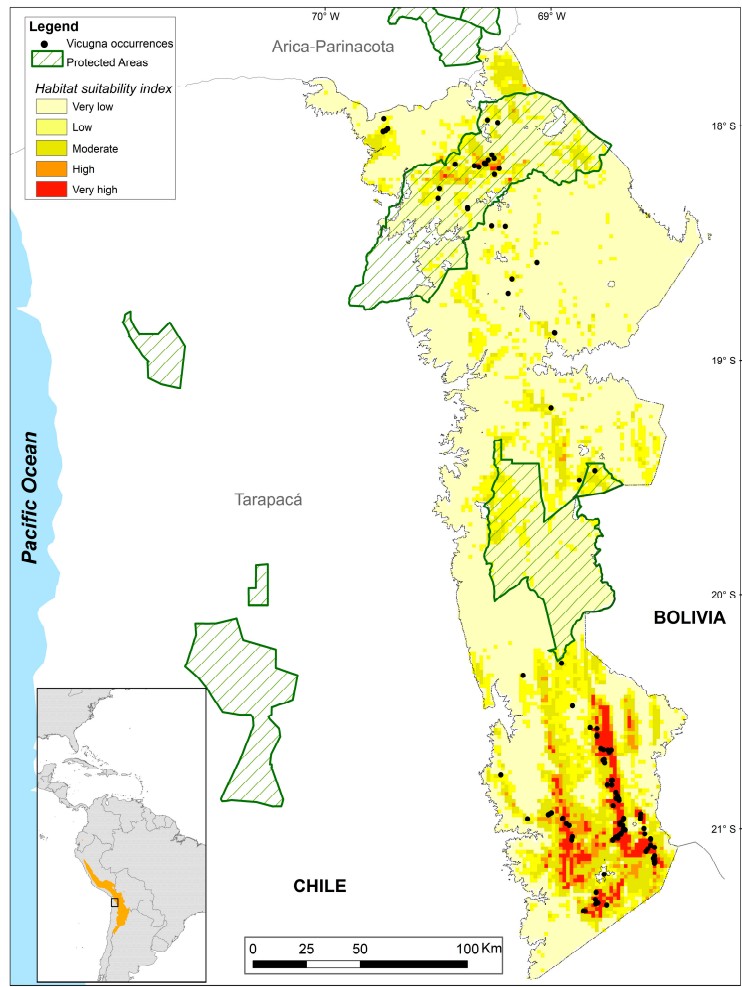

**Figure 1.** Study area and map of potential distribution for vicuñas according to the MaxEnt model. The figure on the right shows the point-wise mean of the 10 output grids with warmer colors identifying better areas for the species presence. Presences of *Vicugna vicugna* detected during sampling of the study area (3700–4900 m asl) are represented by black dots. Chilean Protected Areas are represented with a hatched pattern. In the bottom left inset, the global range distribution of vicugna is marked in orange [33].

### 2.2. Data Collection

The location of vicuñas was recorded in two field campaigns. The first was carried out in April 2012, at the end of the wet season, and the second in November of the same year, in the dry season. Fixed routes were made by car with a total of 1250 km survey in each campaign, with tracks selected to maximize the sampled area of the region. Sampling was thus designed to minimize the biases typical of data repositories for remote areas [34] and allowed the acquisition of data from random locations with the same environmental characteristics. A detailed description of the survey methodology is provided in Malo et al. [24].

### 2.3. Connectivity Analysis

Graph theory was used to model the vicuña distribution landscape following the methodological procedure depicted in Figure 2. The procedure allowed simplification of landscape into nodes and links built on the observed habitat selection [35,36]. First, habitat suitability models for the species (HSMs) were built with MaxEnt at a 1 km grid scale

based on climatic, topographic and vegetation explanatory variables and using our vicuña observations (see more details in Supplementary Materials). HSMs render a numeric value for each grid cell which represents how good it is for the species according to the value of habitat variables present in it, with relevant habitat variables and their loads extracted from the values observed in cells of known presences. Since models for the dry and wet seasons were equivalent, we further worked with one HSM built with vicuña observations irrespective of season. Following HSM building, the nodes or patches of interest for the species in the Tarapacá region were generated applying two cut-off suitability thresholds to the model, so that the core areas for the species (nodes) were defined as those surpassing in the habitat suitability model the value defined as the threshold. One of the thresholds selected was the value that maximizes sensitivity plus specificity (hereafter, MaxSS [37,38]). In short, MaxSS is the model output value for which the balance between the correct classification of actual presences and the erroneous detection of false absences within the training dataset is optimized on mathematical grounds. The second threshold, more conservative, was the average of model values for pixels with recorded vicuña presence (hereafter, AvPP; ref. [39]).

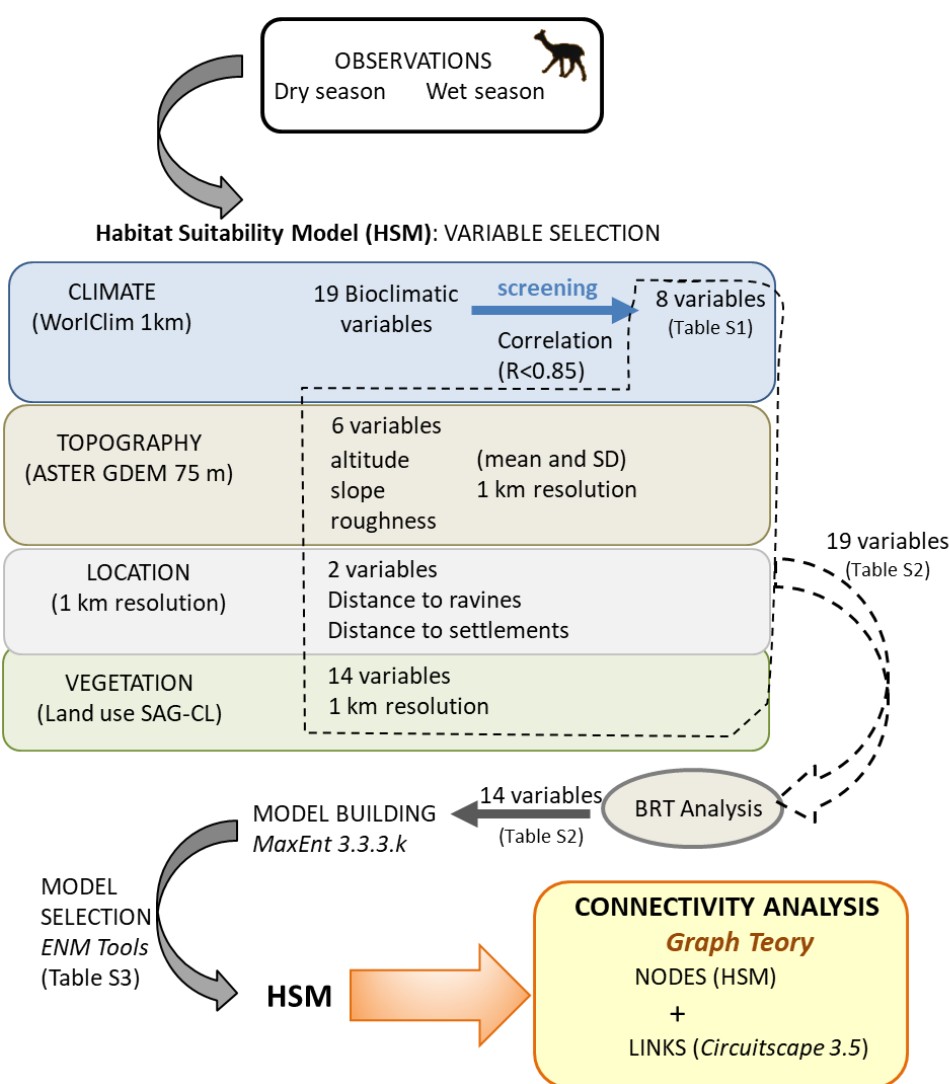

**Figure 2.** Analytical procedure to generate the habitat suitability models (HSMs) and from them to establish (i) the nodes with two different cutoff thresholds and (ii) the links of lower resistance in the landscape according to the dispersive capacity of the species.

The links, or potential best connecting routes between nodes, were computed based on the distance between these patches and the composition of the territorial matrix. To estimate the quality of any connection between two nodes, the resistance to animal movement (or 'friction') posed by each grid cell composing the connection is estimated as the inverse of its HSM value, and the sum of all frictions gives the total resistance of the connector. Circuitscape version 3.5 software [40] was used for this purpose, thus allowing the evaluation of all alternative routes which could connect the patches. To equate the median dispersal distance to an effective resistance value, a linear regression was performed relating the Euclidean (spatial) distance and the effective resistance between nodes (see Panel 2 in Supplementary Materials).

We then calculated the equivalent connected area index [41], which represents the amount of reachable habitat for a given dispersal distance. We computed the normalized equivalent connected area (ECAnorm), a connectivity metric that summarizes the percentage of reachable area of all the nodes available for the species (from the habitat suitability model) compared to the total study area. In the most optimistic connectivity scenario (i.e., probabilities of dispersal equal to 1 among all nodes), the ECAnorm would be equal to the node coverage in the study area.

The dispersal distance for the vicuña has been established at 10.4 km as an average of two approximations: (i) an allometric one following Sutherland et al. [42] and (ii) another based on the home range of the species according to Bowman et al. [43]. For the calculation, the body size of the vicuña was taken from Bonacic [44] and the home range from González et al. [22]:

$$\text{Median dispersal} = 1.45 \times (32.7^{0.54}) = 9.5 \text{ km (Body size} = 32.7 \text{ kg)} \tag{1}$$

$$\text{Median dispersal} = 7 \times (2.6^{0.5}) = 11.3 \text{ km (Home range} = 2.6 \text{ km}^2) \tag{2}$$

The contribution of each habitat patch to overall connectivity was partitioned into the three fractions defined by Saura and Rubio [45]:

$$\text{dPC} = \text{dPCintra} + \text{dPCflux} + \text{dPCconnector} \tag{3}$$

where dPCintra reports the relevance of the amount and quality of available habitat provided by the habitat patch itself (internal connectivity), while dPCflux represents the flow through the connectors from/to the patch as the origin/destination point with the rest. That is, dPCflux quantifies the relevance of connections beginning or ending in a particular node at the scale of the whole area. Finally, dPCconnector reflects the contribution of the patch itself (as a stepping-stone) for the connections between nodes which depend on it. This last fraction becomes more relevant if the patch in question is included in a relevant non-redundant path, as it depends on the existence of alternative paths which do not include the patch under scrutiny. The connectivity analysis was performed using the Conefor Sensinode 2.6. software [46].

## 3. Results

During the two survey campaigns, 111 vicuña sightings were made in the wet season and 110 in the dry season, corresponding to a total of 566 and 464 individuals, respectively.

### 3.1. Habitat Suitability Models

Habitat suitability models were very consistent between seasons reflecting a high degree of overlap for the three statistics provided by the ENMTools 1.4.4 software [47,48] (see Supplementary Materials for additional details), even though only 11.8% of cell grids with sightings coincided between seasons. Therefore, all sightings were grouped for the construction of year-round models, and 149 occurrences were used in the modeling since presences were limited to one occurrence per 1 km$^2$ cell grid.

From the 39 environmental variables considered at the outset (Table S1), the distribution model (MaxEnt) was constructed with 14 variables selected by BRTs (Table S2). Since the predictive performance of MaxEnt models is influenced by the choice of feature types and the regularization constants, models were optimized with a β regularization parameter of 3 and the autofeatures option of MaxEnt (see details in Table S3). The average of the 10 models generated after cross-validation presented a good fit (AUCaverage ± SD = 0.847 ± 0.054), and an average gain of 1.05 for the final model. This indicates that the average likelihood of the presence records is approximately 2.86 times higher than a random background pixel present in the study area.

The HSM for the vicuña reflects suitable areas for the species in the north of the study area, around the 18° S parallel, but they are more relevant to the south, between 21° S and 20° S, because of their larger areas (Figure 1). The distribution of the vicuña according to this model is strongly determined by annual precipitation in the region (Table 1), with the presence of the vicuña being maximized in areas where annual precipitation varies between 60 and 80 mm (Figure S1a). The standard deviation of altitude (a measure of roughness) also influences the presence of this ungulate, increasing notably where it shows low values that reflect landscapes of homogeneous altitude (Figure S1c). Finally, the presence of steppe and open scrub also determines the presence of vicuñas, reaching its maximum in areas where steppe cover is between 30 and 60% (Figure S1d). The jackknife method corroborates the importance of these variables (Table 1), and it also shows the relevance of other climatic variables, such as mean annual temperature and seasonality of temperatures.

**Table 1.** Percent contributions and relative predictive power of different environmental variables in the final vicuña distribution model. Based on the jackknife of regularized training gain in MaxEnt models for vicuña. Values shown are averages over 10 replicate runs. The relevant variables are highlighted in bold.

| Variables | | Contribution (%) | Jackknife Test of Training Gain | |
| --- | --- | --- | --- | --- |
| | | | Only the Variable | Without the Variable |
| Topographic | Mean altitude | 0.57 | 0.014 | 1.048 |
| | SD altitude | **10.34** | **0.168** | **0.938** |
| | Mean gradient | 0.62 | 0.010 | 1.030 |
| Location | Distance to ravines | **11.14** | **0.222** | **0.891** |
| | Distance to settlements | 4.90 | 0.025 | 1.020 |
| Climatic | BIO 1 | 5.10 | 0.013 | **0.974** |
| | BIO2 | 0.20 | 0.062 | 1.051 |
| | BIO 3 | 1.02 | 0.097 | 1.038 |
| | BIO 4 | 5.10 | **0.227** | 1.047 |
| | BIO 7 | 3.02 | 0.064 | 1.052 |
| | BIO 12 | **40.34** | **0.463** | **0.926** |
| Vegetation | Bofedal | 1.16 | 0.002 | 1.039 |
| | Steppe | **9.76** | 0.026 | **0.891** |
| | Very open shrub | **6.72** | **0.129** | 1.010 |

### 3.2. Connectivity Analysis

Nodes were defined from HSMs as areas of greatest habitat suitability for the species through two thresholds. On the one hand, the threshold that maximizes sensitivity plus specificity (MaxSS) was 0.336, and based on it, a total of 111 nodes of interest for the vicuña were delimited, covering an area of 176,600 hectares (Figure 3, left). The links between them had an average effective resistance of 15.70 (range: 0.29–43.07), while the average Euclidean distance was 80.27 km (range: 1.00–228.12 km). On the other hand, the threshold that averages the values of all prediction pixels (AvPP) was 0.568, and this more restrictive threshold yielded 48 nodes with a total area of 64,800 ha (Figure 3, right). The connectors between them presented a range of effective resistances that varied between 0.31 and 32.37,

with an average of 14.46. A graphical representation of connections between nodes for both threshold values can be found in Figure S2. The Euclidean distance between these nodes ranged from 1 km to 227.41 km, with the average distance between them being 93.46 km.

The degree of connectivity measured by the equivalent connected area (ECA) index ranged between $13.59 \times 10^8$ ECA units for the MaxSS threshold and the value obtained for the most conservative threshold that averages the pixel predictions of $5.82 \times 10^8$ ECA units. These features imply that the percentage of reachable area (ECAnorm) for the species lies between 7.7 and 9.0% of the good habitat patches or nodes depending on the defined threshold (Table 2). Furthermore, the breakdown of connectivity into its three fractions for both thresholds shows that the PCintra component is the most important, followed by PCflux (Table 2), while PCconnector presents a very low, almost negligible value.

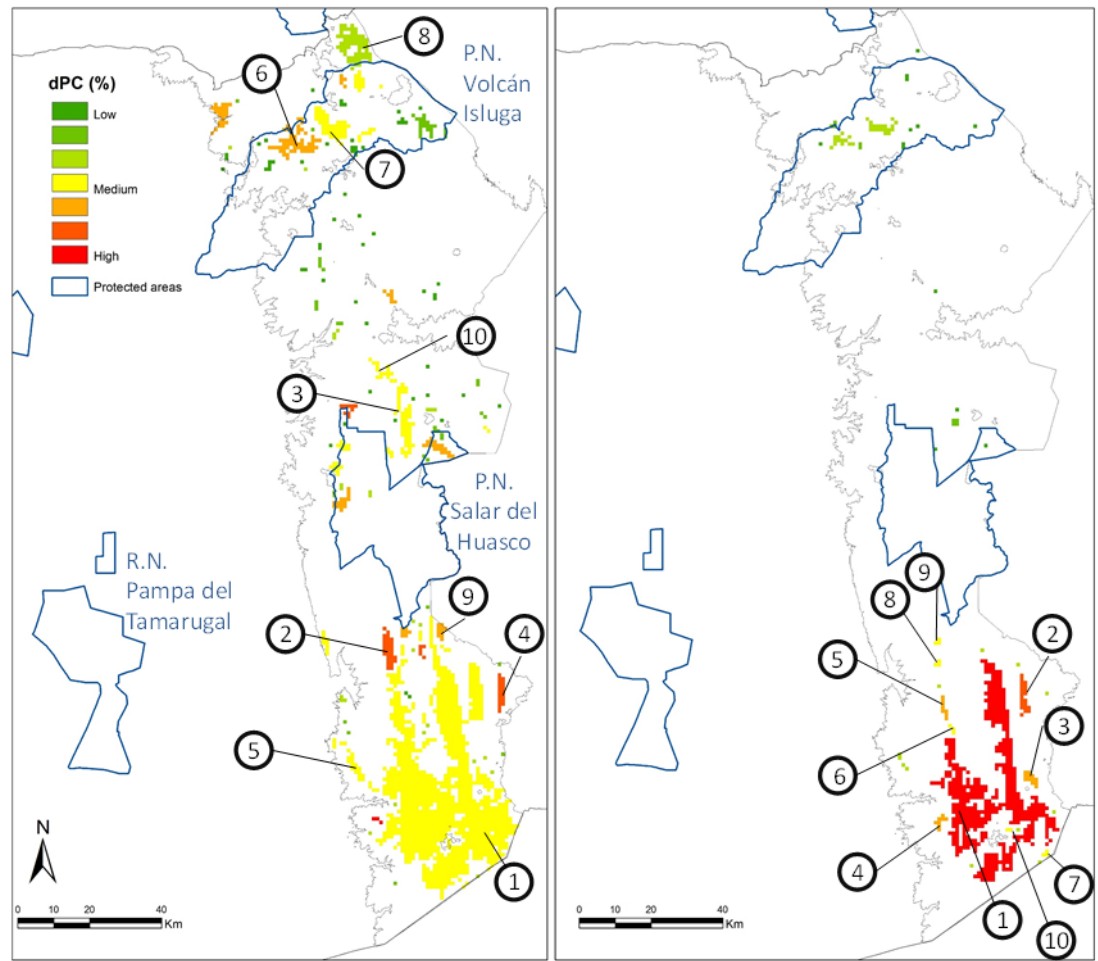

**Figure 3.** Relevant nodes for the vicuña in the Tarapacá region defined according to the two thresholds: MaxSS (**left**) and AvPP (**right**). The relative importance of each node with respect to global connectivity (dPC) in terms of conservation value due to habitat quality for the species and relevance for the connection is represented in a color gradient. The 10 most relevant nodes for vicuña connectivity are identified in each panel (numbered from 1 to 10 in each panel, see Table 3).

**Table 2.** Relative value of pixels by habitat quality and connectivity relevance for the two established cutoff thresholds. Values shown for each dPC fraction are relative contributions (%) to the total importance of individual landscape patches for habitat availability and connectivity in the territory.

| Threshold | ECA$_{norm}$ | dPC$_{intra}$ | dPC$_{flux}$ | dPC$_{connector}$ |
|-----------|--------------|---------------|--------------|-------------------|
| MaxSS     | 7.70         | 55.99         | 43.44        | 0.57              |
| AvPP      | 8.98         | 66.5          | 33.32        | 0.18              |

The spatial representation of the nodes according to the connectivity index allows for the identification of the most relevant habitat patches for the maintenance of connectivity (Figure 3). Node 1, located to the south of the study area, stands out among them for all its parameters. Regardless of the threshold used, this is the largest node (1146 Ha MaxSS threshold and 519 Ha AvPP threshold) and has a high weight regarding the internal connectivity (PCintra component) and as a potential origin or destination of movements (PCflux, Table 3).

The connectivity analysis based on the more conservative AvPP threshold locates the 10 main nodes for the connectivity of the species to the south of the region, and except for the aforementioned node, the rest have an area size of interest between 2 and 21 hectares (Table 3). Complementarily, the scenario based on the MaxSS threshold reflects the presence of optimal nodes for the species throughout the region, from north to south (Figure 3). Among the nodes of the MaxSS scenario, it is worth noting the potential contribution of nodes 3 and 10 to the connectivity between the southern and northern patches (PCconnector component) as connecting elements or bridging tesserae located between 19°5′ S and 20°5′ S. (Figure 3). It is also worth noting the potential contribution to this north–south connection of a series of smaller patches located on the western boundary of the Salar del Huasco National Park (Figure 3). Although these patches are located at approximately 36 km in a straight line from the closest tesserae to the south of the region, no other tesserae of interest for the connectivity of the species have been recorded between 20°10′ S and 20°25′ S (for more details, see Supplementary Materials).

**Table 3.** Relative value of the main nodes according to the quality of their habitat and their interest in connectivity according to the two cutoff thresholds used in this study. The contribution of the node is shown for each of the connectivity fractions, as well as the total area of the node in hectares. The spatial location for each node can be found in Figure 3 with the same numbering.

| Threshold | Node | dPC | $dPC_{intra}$ | $dPC_{flux}$ | $dPC_{connector}$ | Area (Ha) |
|---|---|---|---|---|---|---|
| MaxSS | 1 | 93.833 | 71.140 | 22.511 | 0.182 | 1146 |
| | 2 | 4.019 | 0.046 | 3.943 | 0.031 | 29 |
| | 3 | 3.113 | 0.135 | 2.698 | 0.280 | 50 |
| | 4 | 2.331 | 0.020 | 2.312 | 0.000 | 19 |
| | 5 | 2.049 | 0.016 | 2.007 | 0.026 | 17 |
| | 6 | 1.627 | 0.176 | 1.431 | 0.019 | 57 |
| | 7 | 1.410 | 0.125 | 1.279 | 0.006 | 48 |
| | 8 | 1.405 | 0.236 | 1.169 | 0.000 | 66 |
| | 9 | 1.351 | 0.005 | 1.346 | 0.000 | 10 |
| | 10 | 1.034 | 0.014 | 0.845 | 0.174 | 16 |
| AvPP | 1 | 98.589 | 79.569 | 18.811 | 0.208 | 519 |
| | 2 | 6.165 | 0.130 | 6.026 | 0.009 | 21 |
| | 3 | 3.627 | 0.043 | 3.585 | 0.000 | 12 |
| | 4 | 2.464 | 0.019 | 2.445 | 0.000 | 8 |
| | 5 | 2.239 | 0.019 | 2.220 | 0.000 | 8 |
| | 6 | 0.878 | 0.003 | 0.876 | 0.000 | 3 |
| | 7 | 0.874 | 0.003 | 0.871 | 0.000 | 3 |
| | 8 | 0.698 | 0.003 | 0.696 | 0.000 | 3 |
| | 9 | 0.650 | 0.003 | 0.647 | 0.000 | 3 |
| | 10 | 0.608 | 0.001 | 0.607 | 0.000 | 2 |

## 4. Discussion

Our results show that the current ecological connectivity of the vicuña in the Tarapacá region is limited by the suitability of the habitat after a marked global and historical population reduction that affected the species. The high-altitude environment inhabited by the vicuña is not homogeneous throughout the study area, so short-term connectivity would depend on the movement of individuals between nearby, though frequently distant, habitat patches. This happened even though a wide home range was used in

this study [22], but smaller dimensions have been reported in daily, seasonal and annual movements of the vicuña [21,23], and they would lead to further reductions in connectivity if considered. However, we suspect that this effect is offset by potential mid-term dispersal and long-term gene flow [49]. Although there are no data on movements of dispersing juveniles [27], genetic influence at the population scale has been reported to reach hundreds of kilometers [50]. The difficulties stated here in fixing a potential dispersal distance for the species reflect the complexities of mathematical modeling to describe landscape connectivity, and they point to future research demands to make connectivity models more useful for conservation [51].

The variables in the selected model account for a species adapted to living in arid conditions, with low to medium vegetation cover and a rather uniform topography. Thus, our results indicate that the probability of vicuña presence increases around 50 mm of annual precipitation with a maximum of between 60 and 80 mm. Such little rainfall occurs mainly during the summer, increasing soil moisture and causing synchronous vegetation growth in all plant communities [32]. Synchronicity of vegetation probably explains why vicuña groups do not show large differences in spatial patterns between the dry or wet seasons [24], so vicuña groups are found in zonal vegetation zones of the steppe with a plant cover between 30 and 60% and open scrub. Also, the probability of presence increases in places far from ravines with temporal streams, with a maximum of between 10 and 15 km away from them, and in areas with low mean standard deviation of altitude. Since sampling was carried out from dirt roads which circumvent very steep areas, the selection of flat areas by the vicuña could be somewhat overrated in our data, but the bias is probably small as such selectivity fits with the habitat preference traditionally described for the species [27,29].

Contrary to our expectations, the percentage of vegas and bofedales showed a low importance score to explain the presence of the vicuña. However, these small wetlands are used and even preferred by vicuña groups at smaller spatial scales due to the higher nutritional quality of the plant communities in them [52,53] compared to plant species found in grassland and scrublands, which are also consumed [28,54]. Such low importance in the model could be associated with the low frequency of meadows and wetlands in comparison to other environments, as they cover only 1.1% of the regional surface [55]. Most vicuña records occurred in grid cells dominated by zonal vegetation, but the dimensions of most wetlands in the region could be too small to be properly reflected in the pixel size of the landcover layer used as the basis for the modeling. Therefore, it is expected that these results regarding the use of bofedales and vegas would change at smaller scales as reported at the population scale in our study area [24] or through the tracking of individuals in their daily movements in other study sites [53].

Our results indicate that habitat suitability is very heterogeneous in the Tarapacá region even at a 1 km grid cell scale, and such heterogeneity is reflected in the potential distribution and dispersal of the species. The highest probability values of vicuña presence are concentrated in the north and south of the region, with a low probability area in the center due to the environmental factors included in the model. In fact, in the central zone of the study area is located the Tarapacá ravine, which has been traditionally considered a geographic barrier for the species due to its depth and abrupt topography of its surroundings.

*Conservation Implications*

Globally, the vicuña is classified as of Least Concern according to the IUCN RedList [20]. However, following the methodology proposed to assess biodiversity at the country level [56], the vicuña in Chile has been classified as Vulnerable with a high probability of recolonization in case of local extinctions [57], with local extinction being considered one of the first signs of biodiversity loss [19]. The Tarapacá region holds roughly 2% of the total vicuña population in Chile, and according to our results, its persistence would be improved by maintaining the populations in the north and south of the region as population nuclei

with high levels of internal connectivity. Moreover, the connectivity analysis underscores the relevance of connectors at the regional scale, as shown by the relative values of dPCintra and dPCflux in Table 2 (see below). Despite not considering connectivity to the east, a low contribution of individuals from there is suspected due to the high levels of poaching reported for that area [20]. However, a connectivity analysis of neighboring populations extending to the east of the Andean summits could help direct conservation actions and inform a transboundary, integrated management plan for the species [58].

Current and potential connectivity could help the genetic exchange in an area considered of evolutionary importance due to the hybridization of the two vicuña subspecies. The Tarapacá region shows the presence of genetic hybrids [49] with morphological expression between *V. v mensalis* and *V. v. vicugna* [22]. In this way, connectivity would prevent inbreeding in small population nuclei scattered throughout this desert area, increasing the possibilities of adaptation and evolution of a new diversity in this area of subspecies contact [59].

According to the MaxSS threshold, there are three nodes of potential interest in the extreme north of the region (6, 7 and 8) relatively close to the populations of the protected areas of northern Chile out of the study area (Salar de Surire, Las Vicuñas Natural Reserve and Lauca N.P.), and three intermediate ones (2, 3 and 10) that would facilitate their connection with the large area of southern Tarapacá. These six areas would be the priority sites for action with a regional connectivity conservation perspective, with two of them (6 and 7) inside Isluga Volcano National Park and the others outside protected areas [60]. In all cases, the analysis gives a high relevance for connectivity to these nodes, even though their size and value as habitat patches is low (note the relation between dPCintra and dPCflux values in Table 3).

Our results highlight the relevance of the southern end of the study region, where an area of high connectivity potential (mainly due to the dPCintra component) accompanied by high density is detected [30]. The whole area in the south is outside any protected site [60], and it overlaps in part with areas of intense mining activity [26]. This situation exemplifies one of the main challenges of the 21st century: making activities that entail a high disturbance intensity compatible with conservation objectives to effectively manage biodiversity in areas like mining regions [61]. Beyond the implementation of specific conservation actions, like the declaration of small reserves and the enforcement of management plans within protected areas, conservation of the species in these large areas will depend on proper management of mining sites and their surroundings [61]. The conservation of the species within mining lands may take advantage of the fact that, because of surveillance, poaching is reduced there, a fact which may generate a refuge effect for the species as suggested by Mata et al. [26].

Promoting connectivity within conservation strategies is undoubtedly a key issue, especially in the current context of climate change, with fluctuations in water regimes and rising temperatures. To some extent, even if the long-term effect on different species is unknown, adequate connectivity could help the most mobile species to respond to these changes [62]. Given the IPCC (2023) projections, which estimate a 40% reduction in precipitation and an increase in aridity in the Altiplano of northern Chile [63,64], the subspecies *V. v. vicugna*, better adapted to arid systems, could potentially expand its distribution northward provided connectivity to these areas is increased.

In this context, it is essential that the planning of conservation areas and actions is carried out through formal procedures of habitat modeling and evaluation of the territory from the perspective of nodes and potential connectors, such as those used in this study. Also, complementary methodological approaches or variants (as the MaxEnt thresholds tested here) informing different aspects or objectives of the actions to be implemented can help.

From the perspective of mining and its role in vicuña conservation, it is necessary to consider that the loss and degradation of habitats for the species rank high among the impacts derived from mining activities. The high water demand from mining has an

intense impact on wetlands [65], which are one of the preferred habitats for the species. Also, roadkill is a direct impact derived from this activity due to the intense circulation in mining areas [26], which may, or may not, be compensated by increased survival due to protection from poaching. The total area of mining concessions in operation in the Tarapacá region is about 20,000 km$^2$, which is 49% of the province's surface [66], compared to only 9% of protected areas. In this situation, the co-responsibility of these companies could be an ally in the conservation of vicuñas, and of numerous species with which they share the habitat. Reconciling conservation objectives with other land uses, especially in the case of high-impact activities like mining, and more generally involving local stakeholders to ensure the long-term sustainability of conservation strategies, represents a real and critical challenge [67].

In short, the results of this work can facilitate the long-term viability of vicuña, helping to design actions to promote the structural connectivity of the territory through effective protection within protected areas, internal management in productive areas such as mining, as well as habitat restoration in some sites (i.e., bofedales, ref. [68]). This set of actions can achieve functional connectivity of the vicuña, promoting its movements and genetic flow between north and south. Finally, the methodological exercise presented here for one species in one region could serve as a model to drive the design of connectivity conservation plans for other sites or species, especially for species present in low numbers over vast remote areas, where the application of remote sensing and geospatial tools is a necessity.

**Supplementary Materials:** The following supporting information can be downloaded at: https://www.mdpi.com/article/10.3390/land13040472/s1, Panel 1. Details of model building environmental suitability analysis. Table S1: Comprehensive list of the original variables used to build the habitat suitability models (HSMs). Table S2: Results of variable screening using boosted regression trees (BRT). Table S3: Main results of the model selection process carried out with ENMtools and based on the Akaike Information Criteria (AIC). Figure S1: Response probability of vicugna presence according to the variables included in the model. Panel 2: Computation of distance equivalences. Figure S2. Current map showing potential connection fluxes between core habitat patches for vicuña.

**Author Contributions:** Conceptualization, C.M. and B.A.G.; methodology, C.M.; formal analysis, C.M.; field data N.F.-A., B.A.G., D.S.D. and J.E.M.; data curation, C.M.; writing—original draft preparation, C.M., B.A.G., D.S.D., N.F.-A. and J.E.M.; writing—review and editing, C.M., B.A.G., D.S.D. and J.E.M.; supervision, C.F.E. and J.E.M.; funding acquisition, C.F.E. All authors have read and agreed to the published version of the manuscript.

**Funding:** This research forms part of the dissemination material from the project "Diagnóstico de la ecología poblacional de los ungulados silvestres en la Región de Tarapacá y medidas de solución al conflicto silvoagropecuario—ungulados silvestres", which is supervised by the Tarapacá SAG with funds from Teck-Quebrada, BHP-Billiton and Minera Doña Inés de Collahuasi mining companies. The TEG-UAM research group gets funding from the Comunidad de Madrid and the European Social Fund through the REMEDINAL TE-CM Research Network (P2018/EMT4338). CM was supported by a grant from Santander Bank "Becas Santander Iberoamérica Jóvenes Profesores e Investigadores, 2012".

**Data Availability Statement:** Availability of shapefiles can be requested from the corresponding author, Cristina Mata (cristina.mata@uam.es).

**Acknowledgments:** We thank the Servicio Agrícola Ganadero (SAG) of Tarapacá and the Corporación Nacional Forestal for their institutional assistance, and especially V. Malinarich and J. Valenzuela. I. Nuñez, I. Vasquez, D. Valencia, O. Chacón and C. López assisted with the fieldwork, and P. Acuña and J. Hernández (GEP, Universidad de Chile) with GIS building.

**Conflicts of Interest:** The authors declare no conflicts of interest.

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
