# Peer review of "Ecological Connectivity of Vicuña (Vicugna vicugna) in a Remote Area of Chile and Conservation Implications"

_land, doi:10.3390/land13040472_

Round 1

Reviewer 1 Report

Comments and Suggestions for Authors

The manuscript clearly states the problem and the development of its solution.

In my opinion, the following small details must be corrected, in spite they do not detract from the work that is presented.

In my opinion authors should shortly explain the ecological foundations of the statistics and software used and also many of the terms used, e.g., among others, MaxSS and AVPP, the components of dPC = dPCintra + dPCflux + dPCconnector, etc., which constitute an incomprehensible jargon for the general readers of the journal, which are not supplied only by bibliographic references to them.

It is not clear how it is decided whether a square is a node or a link and how the possible corridors between suitable habitats are defined. It is not clear what ecological characteristics the links have, nor their possible effectiveness.

Perhaps it would be appropriate to indicate the possible corridors on the map in Fig. 3; that is, the spatial pattern of connectivity.

I recommend that the limitations or possible advantages of the method used should be discussed:

1.- Has nothing changed since the year of sampling? Have the vicunas not shown any change in habits or behavior because of the pressures they have received for hundred years? What habitat changes have occurred in the distribution area of ​​the vicuña or its populations and what importance may this have today?

How is the dynamics of the changes from the European conquest to the present day reflected? Are current ecosystems or situations equivalent to those?

2.- Another question: habitats are only internally homogeneous areas? Could the suitability of one area do not depend to a greater extent on internal heterogeneity that optimize and complement the needs of the Vicuñas? What advantages might this broader consideration have in conservation management? Would it be easier to manage these spaces taking into account this heterogeneity? E.g. compatibility with human activities, responses of ecosystems to global change...

Author Response

We are very grateful for the comments, we believe that they have helped us to improve the manuscript a lot. We have included our responses to reviewer 1 at the attached file.

Reviewer 2 Report

Comments and Suggestions for Authors

This is an interesting paper on a species of conservation concern.  It is well-written (though the wording in the discussion degraded a bit, see my suggestions below), the methods are robust, and the conclusions important to conservation. Great job to the author team; I have no major critiques.

Minor suggestions:
37 - on should be in
42 - need to reword, not grammatically correct
50 - this switches from singular to plural, be consistent
76 - delete similar
106 - lower case Vicunas
108 - italicize Vicugna vicugna
110 - replace general situation map with inset
123-124 - I strongly recommend moving some of the information (a summary sentence or two is fine) about the habitat suitability modeling/SDM from the supplmental into the main body for the reader (or reviewer) to know what was done.
Figure 2 - connectivity is spelled wrong
Discussion - talk about how the data points being collected by car might have affected results. Was there a reason that other available points (e.g., from GBIF) were not included?
275 - environment that the vicuna inhabits
288-295 - each sentence starts with "This" - reword  -- and generally, there are a lot of "this" and "these"s in the discussion, see if you can reword to be more concrete
323 - Chile and, according to our results, its...
327 - it would be important -- consider rewording this whole sentence to again get rid of 'this', something like: However, a connectivity analysis of neighboring populations could help direct conservation actions and inform a transboundary, integrated management plan for the species
353-354 - rewrite this sentece, missing its verb
356 - not sure what Betting on means
358 - reword as even if the effect it will have on different species is unknown, adequate...
361 - italicize V. v. vicugna
367 - reword as something like "Also, complementary methodological approaches or variants (as...here) capable of informing different aspects of objectives of the actions to be implemented can help.
371 - degradation of habitats
372 - delete Thus
375 - delete first comma
376 - delete anyhow
378 - delete therefore
380 - reword as "Reconciling conservation objectives with other land uses, and more generally of involving local stakeholders to ensure the long-term sustainability of conservation strategies represents a true, critical challenge [66]."
383 - delete become a tool that (and the s on facilitates)
390 - especially for species

Comments on the Quality of English Language

Discussion just needs a few tweaks to make it flow smoothly.

Author Response

Thank you very much for taking the time to review this manuscript. Please find the detailed responses at the attached file. 
